# Repurposing Established Compounds to Target Pancreatic Cancer Stem Cells (CSCs)

**DOI:** 10.3390/medsci5020014

**Published:** 2017-06-19

**Authors:** Bernhard W. Renz, Jan G. D’Haese, Jens Werner, C. Benedikt Westphalen, Matthias Ilmer

**Affiliations:** 1Ludwig-Maximilians-University (LMU), Department of General, Visceral, and Transplantation Surgery, Hospital of the LMU Munich, Munich 81377, Germany; Bernhard.Renz@med.uni-muenchen.de (B.W.R.); Jan.DHaese@med.uni-muenchen.de (J.G.D.); Jens.Werner@med.uni-muenchen.de (J.W.); 2Ludwig-Maximilians-University (LMU), Department of Internal Medicine III, Hospital of the University of Munich, Munich 81377, Germany; Christoph_Benedikt.Westphalen@med.uni-muenchen.de

**Keywords:** pancreatic ductal adenocarcinoma, cancer stem cells, repurposing, therapeutics, metabolism

## Abstract

The diagnosis of pancreatic ductal adenocarcinoma (PDAC) carries a dismal prognosis, in particular, when patients present with unresectable disease. While significant progress has been made in understanding the biology of PDAC, this knowledge has not translated into a clear clinical benefit and current chemotherapeutic strategies only offer a modest improvement in overall survival. Accordingly, novel approaches are desperately needed. One hypothesis that could—at least in part—explain the desolate response of PDAC to chemotherapy is the so-called cancer stem cell (CSC) concept, which attributes specific traits, such as chemoresistance, metastatic potential and a distinct metabolism to a small cellular subpopulation of the whole tumor. At the same time, however, some of these attributes could make CSCs more permissive for novel therapeutic strategies with compounds that are already in clinical use. Most recently, several publications have tried to enlighten the field with the idea of repurposing established drugs for antineoplastic use. As such, recycling drugs could present an intriguing and fast-track method with new therapeutic paradigms in anti-cancer and anti-CSC treatments. Here, we aim to summarize important aspects and novel findings of this emerging field.

## 1. Introduction

Pancreatic ductal adenocarcinoma (PDAC) is the most frequent malignant tumor of the pancreas. At the time of diagnosis, the majority of PDACs are unresectable and, hence, owing to the lack of efficient systemic therapy, the diagnosis confronts the patient with a dismal prognosis. PDAC has a median survival of just six months and will become the second leading cause of cancer-related death in the U.S. and Europe by 2030 [1,2,3]. In contrast to the steady increase in survival for other gastrointestinal cancers, advances have been slow for PDAC, with a current five-year relative survival of 8–9% [4]. These features can be explained, in part, by the comparatively low susceptibility of PDAC to standard chemotherapeutic treatments, such as gemcitabine (GEM) and GEM-based combinations as well as intensive regimens such as FOLFIRINOX, which, in comparison to GEM, only extend the mean patient survival in stage IV patients from 6.8 to 11.1 months [5]. Furthermore, the majority of patients is diagnosed with advanced-stage disease, either metastatic (50%) or locally advanced cancer (30%), and thus do not qualify for potentially curative resection [6,7]. Hence, novel approaches for therapeutics as well as diagnostics are needed and being currently investigated. Cancer stem cells (CSCs) in general possess outstanding abilities ascribed to stemness—self-renewal and differentiation into heterogeneous cancer cells—allowing them to also act as seeds for relapse and spread within the body of the patient. The highly versatile subgroup of CSCs is located at the hierarchical apex of heterogeneous tumors [8]. Evidence suggests that functionally, CSCs possess the capability to resist conventional treatment and while these can sometimes successfully shrink tumors, a chemoresistant population of CSCs may still survive. Mechanistically, several subcellular means are being discussed and outlined here. First, diverging activity of drug transporters, for instance high aldehyde dehydrogenase (ALDH) activity [9] or low expression of the human equilibrative nucleoside transporters (ENTs) and human concentrative nucleoside transporters (CNTs) which mediate uptake of gemcitabine into cells might provide the necessary tools for chemoresistance in (autofluorescent) CSC-enriched cells of pancreatic origin to conventional therapy [10,11]. Second, low proteasome activity could provide the equipment for drug-resistance in CSCs, especially against newer therapeutic approaches such as bortezomib [12]. In comparison to the bulk of the tumor, CSCs are also defined by additional capabilities, i.e. differing metabolic features. Energy supply in pancreatic CSCs is accomplished mainly by oxidative phosphorylation (OXPHOS), whereas in non-CSCs, glycolysis (Warburg effect) seems to be the predominant metabolic energy source [13]. Activation of autophagy might guarantee sufficient nutrient supply in CSCs that entered a quiescent state or circulate in the blood stream to distant organs [14,15]. Quiescence protects CSCs from chemotherapeutic treatment, which usually targets rapidly proliferating cells [16,17]. Intrinsic or acquired drug-resistance therefore contributes to therapeutic failure of established regimens [16,18]. Mechanistically, acquaintance of drug-resistance is frequently associated with epithelial-mesenchymal transition (EMT), a well-conserved mechanism found physiologically in embryonic development, but also in pathologic conditions, such as fibrosis [19] or cancer [20]. By activation of EMT in cancer, immobile epithelial cells lose their cell–cell adhesions and transform into migratory, spindle-shaped mesenchymal cells that can start the metastatic cascade by migration into adjacent tissues and subsequent invasion into the blood stream [21]. In the case of PDAC, this finely tuned process is under the control of the transcription factor Zeb1 which in turn represses the cell–cell adhesion molecule E-Cadherin via influencing the miR200 family [22]. A strong body of evidence indicates that certain surface markers, such as cluster of differentiation (CD)24, CD44, CD133, C-X-C chemokine receptor type 4 (CXCR4) or doublecortin like kinase 1 (DCLK1), are able to identify a (metastatic) CSC-enriched population in PDAC [23,24,25,26]. Several of these markers have been validated in the past and are in extensive use in enrichment assays for established and primary PDAC CSCs (Table 1). Unfortunately, these markers often have no functional relevance for cancer stemness and are not CSC-specific being also expressed in normal stem cells or tissues. Accordingly, applying these markers to target CSCs is challenging due to their low specificity and lack of potential efficacy. CSC signaling activity or overactivation of pathways such as Wnt/RSPO (R-spondin) [27], c-Jun N-terminal protein kinase (JNK) [28], Nodal/Activin [29], Notch [30] or Hedgehog [24] as well as adaptability of CSCs to the ever-changing tumor microenvironment and CSC niche [31,32,33] are further critical components of the CSC concept. 

Hence, functional CSC qualities that confer the importance of this subgroup within the tumor might not only define their position at the apex of the neoplastic hierarchy, but could also offer the chance to develop CSC-targeted compounds. When compared to non-CSCs, properties such as self-renewal and multipotency, metabolic exclusivity, autophagy, and anoikis emerge as potential therapeutic targets [34]. Consequently, numerous research efforts are underway to specifically target CSCs. Here, we aim to analyze, discuss, and summarize novel treatment approaches with a special focus on previously FDA-approved drugs and their potential to modulate cancer stemness.

## 2. CSC-Specific Treatment Approaches: Preclinical Setting

As mentioned above, multiple qualities determine the special roles of CSCs in liquid and solid tumors [37]. These properties not only equip CSCs with unique advantages in the scope of tumor hierarchies, but, at the same time, could also render them more vulnerable, if specifically targeted.

### 2.1. Influence on the Metabolic Profile of CSCs

With regards to their altered metabolism, an array of intriguing agents emerged for the battle against CSCs. Diverse mechanisms of action are potential targets and will be elucidated in the following section.

#### 2.1.1. Ionophores

Ionophores are chemical species that reversibly bind ions and are used as insecticidal, antihelmintic, and anticoccidial drugs. Apart from these applications, certain compounds within this drug group, such as nigericin [38], which binds K^+^, H^+^, and Pb^2+^ ions or salinomycin, which binds K^+^ ions, are successfully applied against bacteria; moreover, their utility as CSC-specific treatments has already been tested. For instance, salinomycin treatment of breast tumor-bearing mice resulted in tumor growth inhibition in vivo and induced differentiation of CSCs into more differentiated tumor cells with epithelial-like morphology, upregulation of the epithelial marker E-Cadherin, and downregulation of the mesenchymal marker Vimentin [39,40]. Mechanistically, these effects might be explained by a strong inhibition of canonical Wnt signaling [39]. Some of these drugs not only acted as CSC-specific substances, but intriguingly also specifically targeted K-ras signaling by impacting the membrane organization of K-ras and hence, self-renewal of CSCs, without affecting H-ras [41]. The authors discussed the possibility that EMT might also be affected through the same mechanism, because K-ras signaling is at the apex of EMT. In consequence, EMT-induced cancer stemness might be reduced by this mechanism. Moreover, oncogenic activation of K-ras in colorectal cancer sits in control of Wnt pathway induction by activating phosphorylation of its critical receptor low-density lipoprotein receptor-related protein 6 (LRP6) [42]. Thus, in conclusion, some of the selected drugs seem to influence cancer stemness on multiple levels including ion binding, oncogene modification, and CSC pathway regulation.

#### 2.1.2. Antibiotics with Inhibitory Effects on the Mitochondria

Emerging evidence suggests that repurposing inhibitors of mitochondrial protein synthesis from its original antimicrobial use could efficiently target a variety of cancer types of liquid or solid origins. In this context, compounds of the tetracycline or glycylcycline group have been tested for their applicability in CSC-specific treatment. For instance, tigecycline was shown to be an effective agent against acute myeloid leukemia (AML) stem cells by inhibition of cytochrome c oxidase 1/2 (MT-CO1/2), which form subunits of the respiratory complex IV in the electron transport chain (ETC) of mitochondria (Figure 1) [43]. Mechanistically, antibiotic substances of the tetracycline class, e.g., doxycycline, bind to the small mitochondrial ribosomes (28S) and substances of the erythromycin class or chloramphenicol selectively bind to the large subunit of the mitochondrial ribosome (39S) thereby blocking mitochondrial protein biogenesis by preventing protein translation. In functional in vitro substitute CSC-assays, sphere-formation was markedly reduced in a vast variety of different solid tumor types including breast, lung, prostate, and PDAC after treatment with antibiotics of the above-mentioned substance classes [44]. Interestingly, the macrolide antibiotic azythromycin in addition to the chemotherapeutic regimen (paclitaxel and cisplatin) showed a positive effect in one-year survival of stage III/IV non-small cell lung cancer (NSCLC) patients in a small study [45]. At this point, it needs to be further investigated whether the observed effects were based on the CSC-specific nature of azythromycin or its anti-inflammatory effect, especially, since the study was initially designed to monitor cancer-associated infections. Nevertheless, these results propitiate the notion that inhibition of mitochondrial protein synthesis could have an effect on cancer cells and CSCs in particular.

#### 2.1.3. Modulators of Metabolic Functions

Metformin has been investigated extensively for its use as an anticancer compound; for instance, in pancreatic CSCs, it was shown that metformin diminished CSC properties, increased mitochondrial reactive oxygen species (mROS), and induced apoptosis whereas non-CSCs underwent only cell cycle arrest [46]. The same group later showed, that due to their low metabolic plasticity and dependence on OXPHOS, pancreatic CSCs were amenable to metformin-mediated mitochondrial inhibition [13]. Wheaton et al. reported that metformin use affected mitochondrial complex I in colorectal cancer cells [47]. Contrary to the very promising preclinical data, however, the efficacy of the compound in patients was rather inconclusive and in a randomized phase II trial with metastatic PDAC did not show any additional therapeutic benefit [48]. Reasons for the unexpected outcome are multifaceted. In their pioneering work, Sancho et al. reported that CSCs developed resistance to metformin due to an “intermediate glycolytic/respiratory phenotype”. This phenomenon could be treated—and even prevented—by additional MYC inhibition [13]. Moreover, the bioavailability of metformin is only at 50–60% and this is further complicated by the fact that most experimental data are generated by using supra physiological concentrations (mmol/L), whereas physiologically, patient serum concentrations usually only reach the μmol/L range [49]. As already mentioned, tigecycline, on the other hand was able to inhibit respiratory complexes I and IV in CSCs of acute myeloid leukemia presumably by affecting protein synthesis of cytochrome c oxidase I and II (MT-CO1 and 2) without any influence on mROS [43]. The latter reported effects seem to occur mainly functionally in contrast to other agents that actually uncouple and inhibit the respiratory chain, respectively. Atovaquone, an FDA-approved drug for the treatment of malaria, also acts as an OXPHOS inhibitor and was recently shown to target mitochondrial complex III in breast CSCs [50]. Sphere-formation in breast CSCs was significantly inhibited upon atovaquone treatment along with increased mROS levels, whereas normal fibroblasts remained unaffected by this compound. In colorectal cancer cells, atovaquone eradicated hypoxia from CSC-enriched spheres making them more susceptible to radiotherapy adding another interesting angle to this treatment concept [51]. Further anti-malarial compounds have been tested in their utility to modulate metabolic behavior and use as anti-cancer strategy. In this regard, chloroquine, for instance, was shown to have significant effects on pancreatic CSCs due to inhibition of CXCL12/CXCR4 signaling [52] and potentially by suppressing autophagy and metabolic addiction of PDAC [53]. 

In conclusion, FDA-approved drugs may contribute to fight CSCs by inhibiting the mitochondrial respiratory complex modulating their unique metabolic (CSC) attributes, such as OXPHOS. Influencing those behaviors might then lead to a re-differentiation of CSCs to non-CSCs or even to apoptotic events killing the root of cancer. A great p of the mentioned drugs seems to possess a good efficacy in CSC-treatments with most of the knowledge acquired in preclinical setups. However, selected compounds (e.g., chloroquine) are already being tested in clinical trials (Table 2). This process is surely facilitated by the existence of a previous approval through the FDA for a different application.

### 2.2. CSC Pathway Inhibition

CSCs are not necessarily a predefined group of cells. A growing body of literature suggests that CSCs are endowed with a higher grade of plasticity. In this regard, non-CSCs can potentially revert to CSCs presumably by intrinsic or extrinsic activation of cancer stemness-associated signaling pathways such as Wnt/RSPO [27,64], Nodal/Activin [29] coupled with tumor growth factor beta (TGF-β) and the miR-17-92 cluster [16] or Notch signaling. The latter one as well as most of the other mentioned pathways can control EMT and are therefore very potent CSC-inducing mechanisms in PDAC [30]. Interestingly, certain feedback loops exist between the reported signaling pathways; e.g., the miR-17-92 cluster can affect p38alpha and consecutively canonical Wnt signaling in lung cancer [65] suggesting a highly orchestrated and controlled interplay even in deregulated cancer and its CSCs. In the following, we will summarize notable aspects of CSC pathway manipulation by previously established FDA-approved drugs.

#### 2.2.1. CSC Pathway Modulation

Canonical Wnt signaling including its enhancer complex RSPO/LGR (Leucine-rich repeat-containing G-protein coupled receptor) comprises a very intricate pathway system with multiple members on all levels. It is of critical importance in CSCs, particularly in colorectal [66,67] and liver cancer [68], because these neoplasms are inherently Wnt-addicted through activating mutations of the adenomatous polyposis coli (APC) and beta-Catenin genes. By virtue of its complex structure, inhibition of the pathway is not trivial, but various—especially preclinical—attempts have been reported; for instance, salinomycin downregulated Wnt signaling in CSCs of chronic lymphatic leukemia (CLL) [39]. Aprepitant, an FDA-approved small molecule inhibitor directed against the neurokinin 1 receptor (NK1R) that is in use for chemotherapy-induced nausea and vomiting (CINV), intriguingly showed robust efficacy in inhibiting pediatric liver cancer (hepatoblastoma) in vitro and in vivo [54]. More importantly though, we were able to show that aprepitant and other NK1-R inhibitors decreased cancer stemness in colon cancer [55] and hepatoblastoma, most likely by its potent suppressive impact on both AKT/mTOR and canonical Wnt signaling [56]. The latter findings were conceivably conveyed by forcing a disruption of the known CSC transcription factor FoxM1 from its complex with beta-Catenin [69]. Compellingly, FoxM1 is also often dysregulated in PDAC and regulates tumor growth via oxidative glycolysis (Warburg effect) effects [70]. Recently, Blaj et al. showed that low-dose ketamine induced activity-dependent neuroprotector homeobox (ADNP) expression in a colorectal cancer model. ADNP in turn acted as a Wnt repressor which ultimately led to tumor growth inhibition and prolonged survival of tumor bearing animals [60].

Another interesting compound that emerged to be helpful as an anticancer drug in the preclinical setting is disulfiram, known as a remedy to treat chronic alcoholism. Different mechanisms associated with disulfiram treatment seem to target CSCs; in PDAC cells, Kim et al. showed that disulfiram was able to eliminate the highly tumorigenic and therapy-resistant population of Aldefluor^positive^ cells in vitro [58]. Furthermore, in breast cancer cells, EMT was significantly reduced by inhibition of ERK and NFκB signaling [59].

Quinomycin A, a quinoxaline antibiotic, was shown to inhibit spheroid-formation of PDAC and downregulated several CSC markers in the treated group. In vivo, quinomycin A treatment led to a significantly reduced tumor growth and decreased the expression of multiple members of the Notch pathway suggesting that quinomycin A could be of use as a potent CSC-drug in PDAC [62]. Last, crocetinic acid, a compound of saffron, was shown to inhibit pancreatic CSCs by hedgehog inhibition [57] and, as already mentioned, chloroquine inhibited CXCL12/CXCR4 signaling, an important pathway especially for metastasizing pancreatic CSCs [52].

Alterations of DNA methylation are regarded as a pivotal epigenetic mechanism in cancer development. In addition, it has been shown recently that pancreatic CSCs are hypermethylated compared to their non-CSC counterparts. This effect was attributed to increased expression levels of the DNA (cytosine-5)-methyltransferase 1 (DNMT1), which contributed to higher in vivo tumorigenesis as well as CSC self-renewal in PDAC cells. Zebularine, a cytidine analog and inhibitor of DNMT1, afflicted these phenomena and targeted CSC plasticity in PDAC [71].

Taken together, various established compounds appear to affect different CSC pathways and could thereby exhibit their potential as innovative (additional) cancer compounds.

#### 2.2.2. Inhibiting CSC Plasticity by Modulation of EMT

As previously mentioned, EMT is a highly-conserved mechanism that has been shown to be involved in CSC plasticity of many different tumor types, among others breast and PDAC [22,72]. One could hence postulate that inhibition of EMT might lead to a diminished CSC population and in consequence, to a decreased frequency of tumor relapse and metastasis. In this regard, inhibition of EMT has been pursued; for example, Meidhof et al. were able show in a systematic drug screen that the class I HDAC inhibitor mocetinostat acts as an epigenetic drug and interferes with Zeb1 function, restores miR-203 expression, and thereby represses EMT and stemness properties in PDAC and prostate cancer. Moreover, mocetinostat was able to induce sensitivity to chemotherapy, a further CSC hallmark [61]. PDAC cells in particular seem to depend on EMT mechanisms for resistance against the established chemotherapeutic agent gemcitabine; in this regard, it was shown that inhibition of EMT on the one hand increased the expression of the ENT and CNT nucleoside transporters, but did not alter the invasive or metastatic behavior of PDAC to a significant degree on the other hand suggesting that therapeutic strategies targeting EMT might be of great therapeutic benefit [73].

## 3. Future Directions

Out of the collection of presented compounds, more than 50% have not yet been tested as antineoplastic therapeutics in clinical trials on patients with PDAC (Table 2). To translate more promising preclinical data into meaningful treatment options for patients, valid clinical studies need to be conceptualized, carried out, and analyzed. In addition, these results need to be made available faster for researchers, physicians, and especially affected patients. For instance, in the case of (hydroxy)chloroquine as an autophagy inhibitor and enhancer of gemcitabine effects, several studies are currently recruiting and/or closed, however, with results still pending. To speed up the process of translational research, several mechanisms could be helpful. First, intriguing compounds with negligible side effects should be tested earlier and enhanced transition into clinical trials should be faster. Alternatively, consideration on a case-by-case decision basis for selected patients might contribute to a shorter transition time, too. Second, prioritizing of these compounds for clinical antineoplastic use might be necessary. This could be achieved by further preclinical testing in valid in vitro models such as the use of patient-derived organoids or patient-derived xenografts. Last, analysis of conceivable markers as predictors for potential usefulness of repurposed compounds needs to be promoted (e.g., NK1R) expression for aprepitant treatment).

Drugs with FDA-approval harbor the advantage that—in theory—they can be rapidly translated into clinical use. In addition, they typically have little to no side effects compared with classic chemotherapeutic agents. Furthermore, PDAC patients might benefit immensely, if these repurposed remedies turn out to be directed (e.g., only high expressers of full length- and/or truncated-NK1R receive treatment with the NK1R inhibitor aprepitant [55]). 

## 4. Conclusions

In this review, we have discussed current studies that investigate the antineoplastic potential of various established compounds that are already in clinical use for non-neoplastic diseases and conditions. Pancreatic ductal adenocarcinoma, in particular, is a devastating disease and its real Achilles heel is yet to be found. Many different approaches have been extensively investigated over the last decades to improve outcome and survival of patients with PDAC. In this regard, the CSC concept is still relatively new and postulates that, in heterogeneous tumors, the CSC sits at the top of a hierarchically structured pyramid. Apart from the two essential attributes—self-renewal and differentiation—many additional characteristics have been described, which might portray further CSC-subcategories. Moreover, stem and cancer stem cells often cycle between different states and—as such—change their functional properties. Among these, metabolic as well as signaling pathway dependencies seem to profoundly impact plastic CSCs. Interestingly, as demonstrated in preclinical models of PDAC, many FDA-approved drugs display inhibitory effects to one or some of the above-mentioned CSC-specific capabilities. Naturally, that does not mean that these treatment strategies can necessarily be translated into clinical practice. Nevertheless, with the prospect of facing a growing burden of PDAC-related deaths [1,2,3], challenging PDAC and its CSCs with the mentioned drugs or combinations together with established compounds such as gemcitabine-based combinations or FOLFIRINOX could pave attractive new ways for selected patients. Repurposing FDA-approved drugs might therefore emerge as a substantial opportunity with novel perspectives in the treatment of PDAC and its CSCs.

## 5. Outstanding Questions

∗Are authentic mini-organs or organoids derived from primary tumor specimens valuable for preclinical testing of the outlined strategies?∗Can we identify markers as predictors for the antineoplastic efficacy of any of the above-mentioned drugs?∗Is successful translation of the outlined approaches into beneficial clinical applications with CSC-specific potential feasible in PDAC?

## Figures and Tables

**Figure 1 medsci-05-00014-f001:**
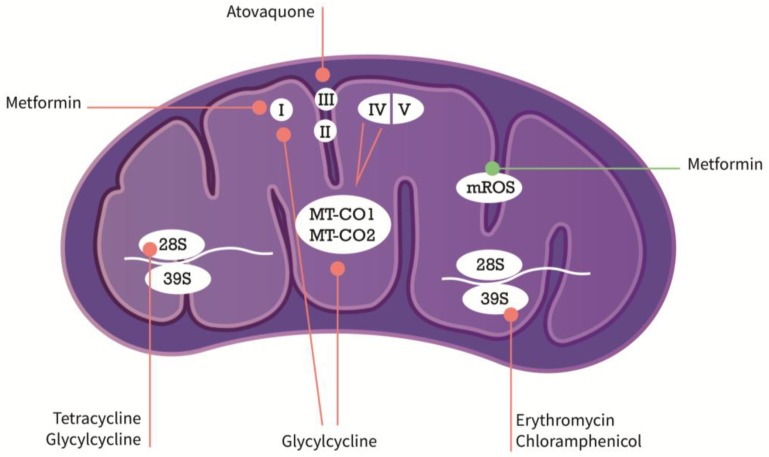
Mechanisms of action of metabolic modifiers in the mitochondria (in purple). Arrows in orange symbolize inhibitory action, arrows in green inducing action. I–V are subunits of the respiratory complex, 28S and 39S mitochondrial ribosomal subunits. MT-CO1/2 are part of respiratory complex IV. For more detailed description, see explanations in the text. MT-CO1/2: cytochrome c oxidase1/2; mROS: mitochondrial reactive oxygen species.

**Table 1 medsci-05-00014-t001:** Markers of cancer stem cells (CSCs) in pancreatic ductal adenocarcinoma (PDAC)**.**

Marker	Reference
Activin	[29]
Aldefluor activity	[35]
Autofluorescence	[10]
c-MET	[36]
CD133	[25]
CD24	[24]
CD44	[24]
CXCR4	[25]
DCLK1	[23,26]
ESA/EpCAM	[24]
Nodal	[29]
Wnt susceptibility	[27]

CD: cluster of differentiation; CXCR4: C-X-C chemokine receptor type 4; DCLK1: doublecortin like kinase 1; EPCAM: epithelial cell adhesion molecule

**Table 2 medsci-05-00014-t002:** Mechanisms, pathways, and clinical trials of compounds for repurposed application in PDAC.

Molecule	Function	Pathway	Cancer Type	Phase/Clinical Trial	NCT Number	Reference
Aprepitant	Anti-emetic	AKT/mTOR, Wnt	Hepatoblastoma, Colorectal cancer, PDAC	N/A	N/A	[54,55,56]
Atovaquone	Anti-malaria	Mitochondrial inhibition	Breast cancer	Phase I	NCT02628080	[50,51]
Azithromycin	Macrolide	Unknown	various: e.g., NSCLC, PDAC	N/A	N/A	[44,45]
Chloroquine	Anti-malaria	CXCL12/CXCR4; Autophagy	PDAC	Phase I	NCT01777477	[52]
Crocetinic acid	Safron compound	Hedgehog inhibition	PDAC	N/A	N/A	[57]
Disulfiram	ALDH inhibitor	ERK/NFκB/EMT	Breast, Colorectal, Melanoma, PDAC	Phase I	NCT02671890	[58,59]
Ketamine	NMDA receptor antagonist	Wnt inhibition	Colorectal cancer	N/A	N/A	[60]
Metformin	Biguanide	Mitochondrial inhibition	PDAC, colorectal	Phase II	multiple: e.g., NCT01210911, NCT01971034	[48]
Mocetinostat	Class I HDAC inhibitor	EMT inhibition	PDAC, prostate cancer	Phase I/II	multiple: e.g., NCT02805660, NCT00372437	[61]
Nigericin	Ionophore	EMT inhibition	Colorectal cancer	N/A	N/A	[38]
Quinomycin A	Quinoxaline antibiotic	Notch inhibition	PDAC	N/A	N/A	[62]
Salinomycin	Ionophore	Wnt inhibition, K-ras	Breast cancer; CLL	N/A	N/A	[41,63]
Tigecycline	Glycylcyclin	Mitochondrial inhibition	AML; Breast, Lung, Prostate, PDAC	Phase I	NCT01332786	[43,44]

ALDH: aldehyde dehydrogenase; AML: acute myeloid leukemia; CLL: chronic lymphatic leukemia; EMT: epithelial-mesenchymal transition; ERK: extracellular signal-regulated kinase; HDAC: histone deacetylase; N/A: not available; NFκB: nuclear factor kappa beta; MDA: N-methyl-D-aspartate; NSCLC: non-small cell lung cancer; PDAC: pancreatic ductal adenocarcinoma; referenced trials were searched on https://clinicaltrials.gov/ct2/home.

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
