# Peer review of "Repurposing Established Compounds to Target Pancreatic Cancer Stem Cells (CSCs)"

_medsci, 2017, doi:10.3390/medsci5020014_

Round 1

Reviewer 1 Report

Metformin - should coment on the bioavailability issue. Sancho et al. also suggested that metformin alone would result in the rapid develpopment ofn resitance. Combinations with Myc inhibiting agents should be discussed. 

Disulfiram - promising preclincal data have been added, should be added. Please also comments about PK issues.  

DNMT1 inhibition has been shown to affect pancreatic CSC, This should be added to the paragraph about Epigentic modulators. The latter should be revised to improve clarity and could also be combined with the 3.2.2. 

In the conclusion section, the authors should try to prioritize compounds for potential clinical translation and what should be done to speed up this process. 

Author Response

The authors thank this reviewer for his comprehensive review and valuable suggestions to improve the manuscript. Please, see the point-by-point letter of response below.

Metformin - should coment on the bioavailability issue. Sancho et al. also suggested that metformin alone would result in the rapid develpopment ofn resitance. Combinations with Myc inhibiting agents should be discussed. 

This is a very good point and we added the discussion of the above-mentioned issues to the manuscript.

Disulfiram - promising preclincal data have been added, should be added. Please also comments about PK issues.  

Use and effects of disulfiram were added to the manuscript as well as Table 2.

DNMT1 inhibition has been shown to affect pancreatic CSC, This should be added to the paragraph about Epigentic modulators. The latter should be revised to improve clarity and could also be combined with the 3.2.2. 

A section concerning DNMT1 and its inhibition was added as suggested by this reviewer. 

In the conclusion section, the authors should try to prioritize compounds for potential clinical translation and what should be done to speed up this process. 

A section concerning DNMT1 and its inhibition was added as suggested by this reviewer. 

Reviewer 2 Report

The review article summarizes the repurposing established compounds to target pancreatic cancer stem cells (CSCs). Despite the advances in molecular pathogenesis, Pancreatic Ductal Adenocarcinoma (PDAC) remains a major unsolved health problem in the world, due to aggressive behavior, highly dense dismoplasia and the poor prognosis. The five year survival rate is less than 8% and a median survival time after diagnosis of less than six months. PDAC is a rapidly invasive, metastatic tumor, which is resistant to standard therapies. At present, nab-paclitaxel with gemcitabine based chemotherapy is the mainstay treatment for metastatic PDAC but none of the agents have objective response rate of over 31%. CSCs are the cells within a tumor that exclusively have self-renewal capacities, can give rise to all cancer cell lineages within a tumor, are exclusively tumorigenic in vivo and have ability to undergo asymmetric/symmetric cell division, enabling CSCs to maintain and expand themselves and also a distinct profile of surface marker expression that has actually been linked to poor prognosis.  CSCs potentially play a critical role in tumor relapse and metastasis. Numerous research efforts are underway to develop therapeutic strategies that target these cells. Current review is focusing on to analyze and discuss novel treatment approaches with previously FDA-approved drugs and their potential to modulate cancer stemness.

Overall, this is well written article, there are few minor has to be addressed:

1.      Author needs to show pancreatic cancer stem markers expression in separate sub heading

 (example 2.1).  

2.      Author did not mention that DCLK1 is pancreatic cancer stem cell marker in the review.

3.      Author needs to show summary table of FDA-approved drugs effect on cancer stem cells and its signaling pathways on pancreatic ductal adenocarcinoma which will benefit the readers immensely.

4.      The authors must check the instruction of authors of the journal: Medical Sciences carefully the cited references and the references section must follow the journal style. 

5.      The authors need to show “future direction” that how Repurposing FDA-approved drugs will benefit in PDAC patients.  

Author Response

The authors thank this reviewer for his comprehensive review and valuable suggestions to improve the manuscript. Please, see the point-by-point letter of response below.

Overall, this is well written article, there are few minor has to be addressed: 

1.      Author needs to show pancreatic cancer stem markers expression in separate sub heading  (example 2.1).  

We answered this by inserting an extra table (Table 1) that lists the surface markers for CSC in PDAC known to the authors including the references for the describing publication.

2.      Author did not mention that DCLK1 is pancreatic cancer stem cell marker in the review. 

DCLK1 was added to the script.

3.      Author needs to show summary table of FDA-approved drugs effect on cancer stem cells and its signaling pathways on pancreatic ductal adenocarcinoma which will benefit the readers immensely.

This is an excellent comment and we appreciate the reviewer mentioning it. We inserted a second table with all the FDA-approved drugs mentioned in the text as well as their known effect on signaling pathways and clinical studies.

4.      The authors must check the instruction of authors of the journal: Medical Sciences carefully the cited references and the references section must follow the journal style.  

Done.

5.      The authors need to show “future direction” that how Repurposing FDA-approved drugs will benefit in PDAC patients.  

A further section “future directions” was added and discusses how we envision the benefits of repurposing drugs to future patient care.

Reviewer 3 Report

This is a well written and eloquent review.

Dr Ilmer is an authority in the field and here provides a candid overview of the literature in this area.

The writing can be improved at several instances, I recommend to go through it again and perhaps request some editorial help.

Author Response

We thank this reviewer for his positive review.